# Back to School: Academic Functioning and Educational Needs among Youth with Acquired Brain Injury

**DOI:** 10.3390/children9091321

**Published:** 2022-08-30

**Authors:** W. Michael Vanderlind, Lauren A. Demers, Georgina Engelson, Rollen C. Fowler, Melissa McCart

**Affiliations:** 1Department of Pediatrics, Oregon Health & Science University, Portland, OR 97239, USA; 2Institute on Development and Disability, Oregon Health & Science University, Portland, OR 97239, USA; 3School of Graduate Psychology, Pacific University, Forest Grove, OR 97116, USA; 4School of Education, Liberty University, Lynchburg, VA 24515, USA; 5The Center for Brain Injury Research and Training, University of Oregon, Eugene, OR 97403, USA

**Keywords:** acquired brain injury, traumatic brain injury, neuropsychological outcomes, academic achievement, school reentry, special education

## Abstract

Youth with a history of traumatic or non-traumatic acquired brain injury are at increased risk for long-lasting cognitive, emotional, behavioral, social, and physical sequelae post-injury. Such sequelae have great potential to negatively impact this population’s academic functioning. Consistently, poorer academic achievement and elevated need for educational supports have been well-documented among youth with a history of acquired brain injury. The current paper reviews the literature on neuropsychological, psychiatric, and academic outcomes of pediatric acquired brain injury. A discussion of special education law as it applies to this patient population, ongoing limitations within the field, and a proposal of solutions are also included.

## 1. Introduction

Brain injuries, including traumatic brain injury (TBI) or non-traumatic brain injury (e.g., stroke, meningitis, anoxic brain injury), are among the most common presenting problems within pediatric critical care settings [1]. In fact, traumatic brain injury is among the leading cause of morbidity and mortality among youth, with prior work documenting nearly 500,000 annual cases of TBI in the United States among newborns to 14-year-olds [2]. Moreover, lifetime estimates of pediatric TBI based on parent report are approximately 2.5%, representing 1.8 million children under the age of 18 years old [3]. Non-traumatic acquired brain injuries are less prevalent than TBI. For example, the yearly incidence of pediatric stroke ranges from 2.5/100,000 [4] to 13/100,000 [5], with an average of 6.7/100,000 when combining ischemic and hemorrhagic stroke rates [6]. Prevalence rates of bacterial meningitis among febrile infants range from 0.46% to 1.2%, with rates decreasing as age increases [7,8]. There are less data on the base rates of pediatric anoxic or hypoxic brain injuries, possibly because of the various causes that result in such injuries. Notably, though, 1 in 301 boys and 1 in 913 girls from birth to 19 years old are hospitalized for near-drowning, a primary cause of these injuries [9].

Children with brain injury are at increased risk for long-term neurocognitive, emotional, social, and behavioral sequelae [1,10,11,12]. Such sequelae place youth at increased risk for poorer educational functioning post-injury [10,13,14]. Indeed, prior research shows that children with a history of brain injury have reduced academic achievement and greater educational needs following school re-entry [15,16,17,18]. Yet, pediatric brain injury is under-identified within school settings [19,20]. Under-identification of pediatric brain injury occurs for multiple reasons, including ineffective communication between medical and education systems as youth return to school. Perhaps more commonly, in the case of mild brain injuries, brain injuries are under-reported or teachers, parents, and clinicians fail to link academic challenge to said brain injury [21].

The current paper aims to review the neuropsychological and psychiatric sequelae of pediatric brain injury and the academic outcomes and educational needs among this patient population. We then provide an overview of special education law, specifically the Individuals with Disabilities Education Improvement Act of 2004 (“IDEA”) and Section 504 of the Rehabilitation Act, that supports pediatric brain injury populations within school settings. Finally, we highlight existing limitations to the identification, evaluation, and provision of educational supports to youth with brain injuries and identify potential next steps.

## 2. Methods

We conducted a narrative or traditional review of neuropsychological, psychiatric, and academic functioning among youth with acquired brain injury. A PubMed and Google Scholar database search was conducted to locate relevant published work. Search terms included pediatric brain injury OR traumatic brain injury OR childhood stroke OR neurological infection OR meningitis OR hypoxic brain injury OR anoxic brain injury AND cognitive OR psychiatric OR academic OR school (re-entry) AND sequelae OR outcomes. Additional literature on special education law was reviewed as well. For select seminal articles, cited references and subsequent articles that cited those works were reviewed (i.e., “snowballing”). The authors reviewed eligible articles and excluded articles that were not relevant to the primary topic, case reports, and non-English language studies. A total of 164 articles were included in this review paper. Authors also provided critical and objective analysis on the current state of limitations to educational supports for youth with brain injuries.

## 3. Literature Review

### 3.1. Sequelae of Childhood Brain Injury

Neuropsychological and psychiatric outcomes following brain injury are largely dependent on the location, nature, and severity of the injury [22]. In addition to specific impairments related to the injured area(s) of the brain, more generalized impairments from widespread trauma, such as inflammation, swelling, and diffuse axonal injury can also occur [23].

#### 3.1.1. Cognitive Sequelae

The cognitive domains that are the most severely disrupted by pediatric brain injury include executive functioning, processing speed, attention, and, to a lesser degree, verbal memory, fluency, and planning and problem solving [24,25,26,27,28,29,30,31,32,33]. Broadly, executive functions are the skills necessary to regulate thoughts, behaviors, and emotions and include skills such as impulse control, self-monitoring, behavioral and emotional modulation, task initiation and completion, problem-solving, and cognitive flexibility. In children with brain injuries, executive difficulties are generally most pronounced in the areas of cognitive flexibility and inhibition [24,34,35]. Critically, due to the protracted nature of brain development, long-term executive functioning sequelae may not become apparent until adolescence [36].

In TBI populations, prior findings have demonstrated a persistent dose-dependent relationship between injury severity and cognitive outcome [37]. Challenges with memory, attention or working memory, and processing speed tend to remit several months to years after injury, though they remain relatively more pronounced and persistent in those with more severe injuries [26,37,38]. Moreover, parents of children with severe TBI continue to perceive executive functioning problems in their children up to three years after their injury [39]. In addition to injury severity, age of injury has been shown to moderate long-term sequelae. Recent findings show that executive functioning challenges are often most devastating in children whose TBI was before age 5 years [37].

Similar to TBI, injury severity and age of injury are predictive of cognitive recovery in childhood stroke. Childhood stroke survivors are at greatest risk for long-term cognitive sequelae if their stroke was more severe [40] or occurred very early in life, particularly between 29 days and six years of age [24,41]. Further, risk is increased if one develops post-stroke seizures [41]. Research has demonstrated deficits in memory [32,33], working memory and attention [40,41], and processing speed [41].

Neurological infections, such as meningitis, can introduce significant risk for developing long-term cognitive deficits and learning difficulties [29,42,43]. Again, greater illness severity and younger age at time of brain injury (i.e., infancy) predict more long-term impairment [43,44,45,46,47]. Further, bacterial infections, relative to viral infections, tend to have more deleterious effects on cognitive development [44]. The most common cognitive changes documented in this population include attention problems and slowed cognitive processing [28,29].

There is relatively less research on long-term effects of anoxic and hypoxic brain injuries in youth, though outcomes are often more severe (i.e., less likely to survive and leave the persistent vegetative state, less likely to become independent in everyday life, and more likely to have seizures of higher frequency) relative to outcomes following TBI [48,49,50]. Still, the extant literature has demonstrated a persistent dose-dependent relationship between duration of hypoxia and cognitive outcome [31,49,51]. The areas of the brain that are more vulnerable to poor cerebral blood diffusion during hypoxia include the hippocampus, insular cortex, and basal ganglia. As such, cognitive challenges in this patient population often include problems with memory, attention, executive function, processing speed, and visuospatial and visuoperceptual functioning [31,52,53,54].

#### 3.1.2. Psychiatric Sequelae

Children with a history of brain injury are at elevated risk for psychiatric disorders for multiple reasons. Brain injuries may impact neural networks (e.g., the limbic system and aspects of the prefrontal cortex including the dorsolateral cortex) that are directly involved in emotion reactivity and emotional and behavioral regulation. Additionally, most children with a history of brain injury report increased levels of life stress following their injury or illness, and stress is a prominent risk factor for a host of psychiatric disorders.

Prevalence of psychiatric sequelae varies by type of brain injury or illness. Whereas 13% of children hospitalized with orthopedic injuries subsequently develop a novel psychiatric disorder, rates range from 36 to 49% in children following TBI [55,56,57], 30 to 59% in children following stroke [58], and 30 to 35% in children following bacterial meningitis [29]. Rates are not well documented for children with history of anoxic and hypoxic events. Across youth affected by brain injury, risk for psychiatric sequelae is highest in those with preexisting psychiatric disorders [59,60], elevated parental distress following injury [12,61], longer PICU stays [62] and delusional memories of their PICU stay [63]. Fortunately, symptom severity often decreases with time from injury [60,64].

Children are at increased risk for externalizing disorders following brain injury. TBI enhances risk for personality change and disruptive behavior disorders [65], with prevalence estimates of oppositional defiant disorder and conduct disorder ranging from 12 to 23% [59]. Most often, there are increases in affective lability, aggression, and disinhibition, and less commonly apathetic or paranoid behaviors [66]. Similarly, parents of school-age survivors of bacterial meningitis report increases in behavioral problems [28]. In comparison, there is less evidence linking externalizing concerns to pediatric stroke survivors [67]. Minimal research has been conducted on externalizing disorders in children following hypoxic or anoxic brain injuries.

Internalizing disorders can also manifest among children with brain injury. Predictors of internalizing disorders following head injuries include premorbid psychosocial adversity and anxiety as well as injury severity [68]. Incidence rates of PTSD are estimated to range from about 6 to 15% in youth with brain injury [58,68,69,70,71] compared to 5% of adolescents in the general population [72]. Incidence rates of novel internalizing disorders vary based on injury type and severity, age of injury, and time of measurement. Following TBI, post-traumatic stress, anxiety, and depression are present in 20 to 25% of children [57,73,74,75,76]. For children with mild TBI, anxiety symptoms generally remit within several months after injury; however, in those with severe TBI, anxiety can become more severe over time [39]. Novel anxiety disorder has also been associated with younger age at time of injury [39,75]. In contrast, older age at time of injury, particularly adolescence, is associated with more depressive symptoms [77]. Of note, though, internalizing disorders remit more often than externalizing disorders among youth with a history of TBI [78]. In childhood stroke survivors, internalizing disorders occur at high rates. Research has shown that within five years post-stroke, 14 to 31% of children meet diagnostic criteria for an anxiety disorder, 24 to 31% for a mood disorder, and 21% for personality change [58,79]. Clinically significant anxiety and depressive symptoms have also been seen shortly after injury in 20–29% of children with history of neural infection [29,80,81]. Moreover, occurrence of psychiatric disorders amongst individuals with histories of bacterial meningitis have been shown to increase to 37% a decade after injury [29]. Minimal research has been conducted on internalizing disorders in children following anoxic brain injuries.

#### 3.1.3. Other Sequelae

Broad social challenges and physical changes are also observed in children with brain injuries. Social challenges include reduced social participation stemming from activity restrictions and changes in functioning [28,82,83,84,85,86], distressing reductions in social acceptance [87,88,89], and changes in peer relationships [90,91,92,93,94]. Generally, those with more severe injuries are more likely to have more interfering and persistent social difficulties [95,96].

With regard to physical changes, it is not uncommon for this patient population to present with motoric and sensory changes related to their injuries. Some degree of novel hearing impairment is especially common (i.e., 30%) in children following bacterial meningitis [29,97]. Other common symptoms associated with pediatric brain injury include headache, nausea, pain, sleep disturbances, and fatigue [29,98,99,100,101,102,103,104].

### 3.2. Academic Outcomes among Youth with Brain Injury

The aforementioned cognitive and psychiatric challenges can greatly influence a child’s school functioning [14,105]. Not only are children absent from school due to hospitalization, rehabilitative care, and ongoing monitoring appointments, but also ongoing symptomatology can impede learning and participation. Difficulties with attention regulation, processing speed, executive functioning, and memory make it harder for children to engage academically and to acquire and retain new knowledge and skill sets. Significant emotional and behavioral challenges can exacerbate cognitive weaknesses, encumber peer and teacher interactions, and reduce academic engagement. Consistently, children with a history of brain injury show poorer academic functioning in a variety of ways, including lower levels of academic achievement and increased need for educational supports.

#### 3.2.1. Academic Achievement

TBI in preschool-aged children is associated with reduced school readiness, which, in turn, is predictive of lower academic achievement later in development [106]. Further, school-aged children with a history of TBI show academic underachievement, though weaknesses may vary across subject area. Word-reading tends to be preserved across TBI especially if the injury occurs after the development of early reading abilities [107,108]. Conversely, TBI has been associated with poorer outcomes on measures of mathematical ability, particularly among those with severe TBI [108]. Severe TBI has also been associated with poorer performance on measures of decoding, reading comprehension, spelling, and arithmetic, when compared to mild and moderate TBI samples, and group differences remained over a five-year follow-up period [15].

Pediatric stroke has also been associated with poorer academic achievement across the domains of reading, mathematics, and spelling [109,110], even when compared to children with orthopedic injuries [111]. Though, there is some evidence that academic underachievement in pediatric stroke may vary across subject areas, with the greatest impairment in mathematics relative to more minor weaknesses in spelling and reading [16,112]. Indeed, in one study, 40% of pediatric stroke patients presented with clinically significant arithmetic weaknesses [16].

Fewer studies examine academic achievement among youth with a history of meningitis or hypoxic or anoxic brain injury. Nevertheless, youth assessed seven to 12 years after bacterial meningitis showed worse outcomes on formal measures of reading, mathematics, and spelling compared to controls, though their functioning tended to fall within the average range nonetheless [113]. Further, parents of children with history of bacterial meningitis reported lower school achievement, higher rates of repeating a year of school, and higher rates of special education referrals six years after infection [28]. Among a small sample of six children with anoxic brain injury, results show academic impairment, though to a lesser extent than intellectual and memory difficulties, following brain injury [114]. One study showed that children with a history of anoxic brain injury have poorer academic outcomes than children with a TBI history [115].

In addition to formal measures of academic achievement, other indicators have been used to examine academic functioning following brain injury. Compared to an orthopedic control group, children with a history of moderate or severe TBI are more likely to be diagnosed with a learning disability and have lower academic competency, per parent or teacher report [17,116]. Concussion, often synonymously referred to as mild TBI, has been associated with greater GPA declines [105], especially for individuals with a history of two or more concussions [117]. Whereas concussion does not appear to impact graduation rates [105], severe childhood TBI has been associated with greater drop-out rates and an increased likelihood of unemployment during adulthood [118]. Teenagers with a history of bacterial meningitis during infancy had lower pass rates on national examinations in England and Wales than those without a history of meningitis [119]. Relatedly, 30% of parents of school-aged children with a history of bacterial meningitis report problems with school achievement [120].

#### 3.2.2. Need for Academic Supports

Paralleling the data on reduced academic achievement among youth with brain injury, there is greater need for educational supports among this population as they begin or re-enter school. Much of the research on school re-entry has focused on pediatric TBI populations. Estimates of the proportion of youth with a TBI history receiving formal supports following their re-entry to school has increased over time. A 2008 study reported 25% of youth with TBI having a formal Individualized Education Plan (IEP) or 504 Plan [121], whereas a 2021 study reported 45% of youth with TBI having an IEP after one year of returning to school [20].

Estimates suggest that nearly 25% of moderate and severe TBI populations are placed in special education classrooms, and nearly 40% repeat a school year or require adaptations [122]. Notably, though, provision of academic supports for less severe TBI groups may increase over time since injury. One study found that youth with severe TBI receive more school supports than mild or moderate TBI groups at a two-year follow up but demonstrated comparable utilization of school services at a six-year follow up [17]. Taken together, academic needs among youth with mild or moderate TBI may increase over time, likely as a function of greater academic demand and emergent executive functioning challenges across grade levels. This trend may also demonstrate the under-recognition of educational needs among less severe TBI during the early stages of returning to school.

Increased rates of academic supports are also documented among non-traumatic forms of acquired brain injury. Pediatric stroke is associated with high rates of special education class placement and provision of academic accommodations; forty percent of youth with a history of arterial ischemic stroke received special education post-injury [18]. Similar data have been reported in a Swedish population-based study, with 38% of arterial ischemic stroke and 24% of hemorrhagic stroke survivors requiring assistance at school [123]. Further, 59% of parents of youth with a history of stroke report that their child needs more help than their peers, and in the same study, 19% of pediatric stroke survivors subsequently attended a special school [124]. Less is known about the provision of educational supports to youth with histories of neurological infection or hypoxic/anoxic brain injury.

#### 3.2.3. School Engagement

To date, there is a paucity of research on school engagement (e.g., participation in classroom instruction) among youth with brain injury. Further, the existing literature on this topic has predominantly come from research on TBI; no studies were identified examining school engagement using samples of youth with non-traumatic acquired brain injuries. Nevertheless, evidence from a classroom observation study shows that children with a history of TBI engage less with academic tasks (e.g., instructed reading, note taking) than do children with a history of orthopedic injury [125]. Moreover, reductions in school engagement, as indexed by absenteeism, have been documented in TBI samples [126]. Students with a history of mild TBI were slower to return to school as compared to a sample of individuals with a history of orthopedic injury [127]. Moreover, higher rates of absenteeism were evident among youth with brain injury relative to extremity injury in the same study.

#### 3.2.4. Moderators of Academic Outcomes

Several factors moderate the degree of academic difficulties following brain injury, including injury severity, degree of cognitive changes, age of injury, and level of premorbid functioning. Greater injury severity tends to predict poorer outcomes, including higher rates of educational supports [17,118]. For instance, in children with history of TBI, there is a negative correlation between initial Glasgow Coma Scale (GCS) score (lower scores indicate greater severity) and provision of academic supports [128]. Similarly, TBI severity is associated with poorer academic achievement [23], higher rates of academic need [116], and greater provision of school supports [128]. However, it is also worth restating that academic needs among youth with a history of less severe TBI may be under-recognized and/or may become more apparent over time [17]. With regard to non-traumatic acquired brain injury, more neurologic complications in the acute stages of bacterial meningitis are linked to poorer academic outcomes, including poorer performance on measures of reading, spelling, and mathematics, across multiple studies [28,115]. Further, youth with a history of moderate and severe birth asphyxia show poorer reading, spelling, and mathematic abilities than those with mild asphyxia or a healthy comparison group [129].

Greater injury-related cognitive change contributes to subsequent academic functioning. In particular, greater levels of executive dysfunction following TBI are associated with weaker academic performance [125]. Relatedly, among a pediatric stroke population, greater parent-rated executive dysfunction was predictive of poorer performance on spelling and, in particular, math measures [16]. Changes in other cognitive domains have been linked to poor academic outcomes. Among youth with TBI, greater language or intellectual impairment is associated with amount of school adaptations [122], greater short-term verbal memory impairment is associated with greater listening comprehension difficulty, and greater long-term verbal memory impairment is associated with greater arithmetic difficulty [130].

Age of injury has also been linked to academic outcomes. Early age of TBI has been linked to flatter trajectories of academic achievement over a five-year span [15]. Similarly, early age of pediatric stroke has been associated with greater academic challenges [111]. Notably, though, one study showed that children who had a stroke between the ages of one and six years showed relative preservations in subsequent academic functioning (i.e., reading and spelling skills) as compared to those who had a stroke earlier (i.e., younger than one year old) or later (i.e., six to 16 years old) in development [110].

Premorbid functioning is a strong predictor of post-injury functioning, especially within the TBI literature. Stronger preinjury educational abilities are predictive of better post-TBI word reading, spelling, arithmetic, and listening comprehension skills [130,131]. Conversely, greater degree of premorbid challenge has been associated with poorer outcomes post-injury. Children with a pre-existing diagnosis of attention deficit hyperactivity disorder (ADHD) or learning disability take longer to return to school following a mild TBI than those without a premorbid neurodevelopmental disorder [132]. Similarly, youth with greater levels of anxiety symptoms prior to sustaining a mild TBI have a more delayed return to school than their relatively lower-anxiety counterparts [132]. A higher number of prior head injuries has also been shown to predict poorer post-TBI outcomes. Number of prior concussions has been positively associated with GPA decline [117] and with more academic dysfunction [126].

### 3.3. Special Education Law

Given the impact of pediatric brain injury on academic functioning, provision of educational supports is paramount. There have been major legislative advances that stipulate the educational rights of these children and document procedures for ensuring these children receive a free and appropriate education (FAPE). Here, we provide an overview of IDEA and the overlapping but distinguishable coverage of Section 504 of the Rehabilitation Act. This review is only intended to touch upon important components of IDEA and Section 504, particularly the referral, identification/eligibility, and placement process. For those seeking a more extended discussion of IDEA, readers are directed to Zirkel [133]. For legal information about Section 504, see Richards [134]. Further information may be obtained from the U.S. Department of Education (https://sites.ed.gov/idea/, accessed on 18 July 2022) and the National Association of State Directors of Special Education (https://www.nasdse.org/, accessed on 18 July 2022).

#### 3.3.1. Individuals with Disabilities Education Act (IDEA)

The overarching purpose of the IDEA is to help states meet the educational needs of students with disabilities through federal funding of state efforts. This charge involves providing students with disabilities a FAPE, including special education and related services to prepare students for further education, employment, and independent living (IDEA, 20 U.S.C. Sec. 1400 (d). There are procedural safeguards designed to allow parental input into a school’s decisions about their child’s education [135]. Eligibility determinations are made on an individual basis by a multidisciplinary team. Only those students whose disability has an adverse impact on their education are eligible to receive special education services through an Individualized Education Plan (IEP). In order to be found eligible for special education services, they must demonstrate need for special education and related services to keep their performance from declining and to ensure they make measurable progress.

Within special education, students with brain injury are provided individually tailored, specially designed instruction that is not only meant to be intensive [136,137], but also applies scientifically validated strategies associated with effective teaching [138]. If a student with a brain injury does not qualify for special education, Section 504 (discussed below) offers coverage to address issues that, while not adversely impacting learning, still substantially limits a major life activity. Section 504 plans may also provide an effective means of monitoring “sleeper effects” associated with brain injury [139].

Within the educational system, parents/guardians and educators may refer a child suspected as having a disability for an evaluation to determine eligibility and need for special education services. IDEA has 13 disability categories that a child may receive services under, with Autism Spectrum Disorder and Traumatic Brain Injury being the last two categories introduced in the 1990 reauthorization of the law. IDEA’s eligibility criteria for TBI includes open or closed head injuries caused by external physical force and excludes eligibility if the brain injury is due to degenerative disease, congenital factors, or birth trauma which adversely affects a child’s educational performance. To date, in most states, youth with a non-traumatic acquired brain injury are not eligible for classification under the TBI classification in IDEA. However, if a non-traumatic acquired brain injury has an adverse impact on educational performance, other disability category options, such as Other Health Impairment (OHI), do exist, so long as the student meets the OHI eligibility criteria and needs special education. If a student does not meet eligibility criteria, it is often because the issues are not adversely impacting educational performance.

Evaluation and identification within special education may occur in two ways. First, a pre-referral team may identify alternative strategies that general education teachers can implement to help a student before a formal referral is made; a referral would be made upon failure of those strategies. The second pathway is the response to intervention (RtI) framework, a multitiered model of prevention using screening and progress monitoring procedures. Students who do not respond to intervention across tiers of progressively more intensive strategies are referred to the multidisciplinary team for special education evaluation [140]. While these models are considered effective practice in public education, they are not applied or used on a wide scale across public school systems [141], with some districts applying their own idiosyncratic approach so long as it still conforms with the procedural requirements of IDEA. Despite their use, pre-referral and RtI models still lack convincing evidence of their effectiveness in correctly identifying students with disabilities [142,143].

#### 3.3.2. Section 504 of 1973 Rehabilitation Act

Section 504 (i.e., 29 U.S.C. Sec. 794) prohibits discrimination against individuals with disabilities in programs and activities that receive federal financial assistance. In public schools this means that students with disabilities cannot be discriminated against by prohibiting participation or providing an inferior education because of their disability; instead, Section 504 is designed to ensure that youth with disabilities receive a FAPE regardless of the severity or nature of the disability. Section 504 covers public schools from preschool through university as well as private schools that receive federal financial assistance. Protection from discrimination extends beyond the provision of education to issues such as architectural accessibility and bullying/harassment by others because of a disability [133]. Section 504 protections are afforded to those who (a) have a physical or mental disability that substantially limits a major life activity; (b) have a record of an impairment or who are regarded as having an impairment; (c) are otherwise qualified individuals with disabilities; and (d) are technically eligible students. Students who are “technically eligible” are those who have a mental or physical impairment that substantially limits a major life activity but do not need a Section 504 plan or services at school (e.g., a student whose impairment is in remission) [134]. Similar to IDEA, Section 504 specifies evaluation and placement procedures, with the purpose being to prevent misclassification and misplacements. Misclassification and/or misplacement often happens to students with a brain injury, which, according to Zirkel [133], has been the focus of more Office of Civil Rights investigations than any other area of Section 504. Finally, Section 504 includes procedural safeguards to protect parents/guardians from schools taking prior action without notice regarding the identification, evaluation, or educational placement of their child with a disability [135].

## 4. Current Status and Future Directions

### 4.1. Current Limitations

A variety of problematic issues exist that can affect acquisition of educational supports among youth with brain injury. Barriers are multifactorial and include communication difficulties between medical, educational, and family systems; challenges obtaining medical documentation of the injury; inappropriate classification categories for non-traumatic acquired brain injuries; minimal involvement of youth within the identification, evaluation, and implementation process; difficulties accessing resources; and educators’ lack of knowledge of brain injury and its sequelae. These current limitations are reviewed in greater detail below.

Many students who experience a brain injury, especially a mild injury, go untreated and/or do not report the event to hospitals or schools [144]. Canto et al. [139] indicated that, across a two-year period at a Florida school district, approximately 1300 school children in its district were treated for TBI at the local hospital but only 129 students were being served by the school district’s TBI program. The findings point to how many students with brain injury—in that study, TBI specifically—are not being screened or identified by school districts for appropriate services under Section 504 or IDEA [144]. Still more students with brain injury are not seen at hospitals nor identified by school districts. These issues point to the fact that many students with milder forms of brain injury are not being identified and are not receiving any form of educational support. Such support may be lacking for the simple reason that school personnel are uninformed about the brain injury and, thus, injury-related sequelae may be mistaken for other high-incidence disabilities such as emotional or behavioral disturbance, specific learning disability, or speech/language impairment, which may misguide the identification of the most appropriate services [139,145,146]. Even in the case of more moderate or severe injuries, there can be problematic communication gaps between medical and educational systems. In one study, over 20% of parents of children with brain injury indicated that school systems did not receive any communication or information regarding the injury [21]. Communication between medical and education sectors tends to be more reliable for youth who are involved in inpatient rehabilitative care or who receive transitional services [21].

Obtaining medical documentation of the injury is essential in the evaluation process for students with TBI. Because TBI is considered a medically related disability such as ADHD, states require formal validation from a provider attesting that a TBI occurred and that it affects educational performance. The primary issue is that often brain injury is not formally documented by medical staff for children with mild brain injury, even if they are taken to an emergency room. Difficulties obtaining such documentation may complicate the evaluation process and may increase the risk for an inappropriate classification. Importantly, though, if a school has good reason to suspect a student’s poor/declining educational performance is due to TBI but overlook it in favor of an “easier,” more expedient, IDEA disability category, it could lead to a FAPE violation and legal proceedings.

Relatedly, a current limitation of IDEA for children with non-traumatic acquired brain injury is the lack of an appropriate IEP classification category. As mentioned, TBI classification is specific to youth who have an acquired change to their cognitive functioning by means of an external force, which precludes children with histories of stroke, meningitis, anoxic brain injury, etc. As such, children with non-traumatic acquired brain injuries are typically classified under a catchment category, OHI, the same category for that is most commonly used for children with a diagnosis of ADHD. While classification categories may seem trivial to the unfamiliar reader, they can affect children’s access to certain types of services. For example, in Oregon, youth with TBI have access to a liaison service that helps to bridge the medical and educational settings by sharing information between sectors, among other types of supports. Conversely, those who present with a non-traumatic acquired brain injury are unable to access such services. Thus, inclusion of non-traumatic acquired brain injury within the TBI classification may equalize the playing field for youth with brain injury within the academic setting.

Evidence suggests that children can effectively achieve self-identified goals [147] and, thus, it may be appropriate to involve a child with brain injury in the development of their IEP goals. In fact, the self-determination movement has encouraged it [148]. However, to date, this movement has not translated from research to the classroom. This problem is attributable to the fact that the IDEA does not require schools to involve students in the development of their IEP; instead, it only requires schools to involve parents to the best of the school’s ability. Despite the empirical support and promise for teaching students with disabilities to identify and develop personalized goals, students’ involvement in the IEP process rarely occurs [148,149]. It is not until age 16 (age 14 in some states) that IDEA requires an IEP transition plan (often referred to as a student-centered planning meeting) be developed. Even then, though, data from the National Longitudinal Transition Study-2 reported that only 12% of students provided any meaningful input in terms of their interests, needs, and preferences or took a leadership role [150,151].

Prescribed accommodations that depend on families having access to certain technology can contribute to educational barriers. In particular, the education system has increasingly utilized technology to help remediate or compensate for areas of weakness. For example, assistive technology, such as speech-to-text devices and communication devices, has been used to compensate for motor or speech weaknesses. Schools also frequently use computer-based software packages to remediate specific learning difficulties. All of these supports, however, depend on having access to the various technologies. As such, reduced access to such technology can preclude families from receiving the services that they qualify for. This issue has become particularly prevalent in the context of distanced learning due to the COVID-19 pandemic, where many families were expected to provide their own technology to support their children’s schooling. Children in families who experience difficulties accessing technology are, in turn, at risk for being under-served.

Finally, limitations to educators’ knowledge on brain injury, best practices for addressing injury-related concerns, and the protocols for recommending evaluations contribute to educational gaps for children with brain injury. It is understood that educators and specialists are extremely busy due to the various federal and state policies and educational reforms they must adhere to. Additionally, teachers wear many “hats” including being case managers, consultants, secretaries, disciplinarians, in addition to collaborating with co-teachers, attending staff meetings and district in-service trainings, managing the ever-increasing amount of paperwork, and communicating with parents on a regular basis [152]. Further, because of the full inclusion movement, and despite lack of training as a disability specialist, general education teachers must now address the severe learning needs of students with disabilities in their classroom [153,154,155]. While principals’ duties have begun to include controlling educational services for students with disabilities, they report possessing insufficient knowledge of special education law as well as dissatisfaction with their administration training in this area [156,157]. Coupled with the lack of building- or district-level resources, the stressful working conditions in many schools have contributed to teachers’ and administrators’ emotional exhaustion. Moreover, these conditions have led many districts across the U.S. to struggle to retain effective teachers or attract individuals to the teaching profession [158,159,160,161]. Combined, these factors (among many more) complicate supporting youth with brain injury in the public-school system. Compounding the challenge is the fact that many administrators, educators, and specialists lack critical knowledge, training, and about brain injury [145,162,163,164]. With these kinds of distractions and burdens placed on school administrators, teachers, and specialists, it should come as no surprise how and why students with brain injury can “fall through the cracks” or not be taken seriously when they do not appear injured.

### 4.2. Avenues for Next Steps

Solutions for ensuring youth with brain injury are properly identified and effectively supported in schools include (but are not limited to): (a) improving hospital to school transition services/models; (b) providing effective professional development and in-service education to administrators and teachers about brain injury and legal responsibilities under IDEA and Section 504; (c) enhancing the IDEA identification process in schools; (d) training parents on how to be more effective in communicating with schools about their child’s educational needs; and (e) changing state-level policies designed to assist schools in identifying and serving students with brain injury more efficiently. Here, we describe some advances that have been made in Oregon, as illustrations of ways to minimize educational gaps for youth with acquired brain injuries.

University of Oregon’s Center for Brain Injury Research and Training (https://cbirt.org/, accessed on 18 July 2022), in collaboration with Oregon’s Department of Education, has created a state-wide TBI Liaison service. This service staffs a group of individuals with expertise in school psychology who partner with school districts to enhance capacity for educating children with brain injury. Their roles are varied but, among them, are working with families to understand, interpret, and share information from the medical sector with the child’s school district. As such, this service highlights transitional services as a mechanism for reducing communication barriers between the medical and educational sectors. Further, liaisons help to educate parents on how to advocate for their children’s educational needs by informing them of their child’s legal rights to a FAPE and navigating them through the proper channels for securing an appropriate evaluation.

An additional pathway to reducing communication barriers between medical and educational sectors is to increase shared knowledge between the two systems. Medical personnel often, with the best intentions, attempt to help children secure appropriate educational supports but are unfamiliar with how special education determinations are made. Seminars for both clinicians (e.g., physicians, rehabilitation therapists, etc.) and education personnel (e.g., teachers, administrators, school counselors and psychologists) may help to address this issue. Within this context, experts from each specialty share the ways in which they interact with youth with brain injury and the types of services that fall under their purview. Using case examples, experts share best practice strategies for how to refer to and consult with each other so as to support a child’s mental, physical, and academic health. Examples illustrate the paths through which communication works best between medical and educational sectors, and a question and answer panel may serve to address ongoing areas of confusion and to debunk myths related to the topic of brain injury from the hospital to the classroom.

A final, crucial avenue for ongoing work is legislative change. Educating parents, educators, and clinicians serves to increase awareness and understanding, but enforceable modifications to educational systems and practices come through the introduction of new laws and the refinement of existing ones. There are many directions for legislative change as it relates to educating youth with brain injury. Among them are the laws surrounding the requirement of medical documentation and the appropriate classification of children with non-traumatic acquired brain injuries. Within the state of Oregon, local education agencies must still document sufficient efforts to obtain appropriate medical documentation; however, should such documentation be unobtainable, youth can still qualify for TBI classification through a “guided credible history interview” process, wherein an interviewer documents one or more TBIs as reported by a credible source and corroborated by another reporter. This legislative change increases access to special education services among families that experience difficulty receiving and paying for medical treatment. Legislative work focused on expanding TBI classification definitions to those with non-traumatic acquired brain injury is currently underway in the state of Oregon. Should a bill be passed, youth with histories of stroke, meningitis, anoxic brain injuries, and more, will gain access to transitionary services, such as CBIRT’s Liaison service, that can help to bridge the gap between the medical and education sectors.

## 5. Conclusions

Understanding of pediatric brain injury and its effect on neuropsychological, psychiatric, and educational functioning is increasing rapidly. Further, the United States has made major legislative advances to ensure that the educational needs of youth with brain injury are being met, and states are continuing to interpret—and modify—how federal law applies to state practices. As illustrated here, there are critical avenues for ongoing work. Educating everyone that interact with youth with brain injuries is essential for best understanding and addressing their needs. In addition to community engagement and education, the importance of legislative advocacy cannot be overstated. Critical examination and revision of educational law is paramount for ensuring that all children—including those with brain injuries—are receiving a FAPE.

## Data Availability

Not applicable.

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
