# Peer review of "Back to School: Academic Functioning and Educational Needs among Youth with Acquired Brain Injury"

_children, 2022, doi:10.3390/children9091321_

Round 1

Reviewer 1 Report

Hello,

The manuscript titled “Back to School: Academic Functioning and Educational Needs Among Youth with Acquired Brain Injury” summarizes the various facets of issues for children with acquired brain injury in educational performance. Academic growth and ongoing support go hand in hand with children who requires special needs. In addition, positive encouragement and identifying barriers is vital. Vanderlind et.al, have reviewed many literatures, identified limitation in the current process and suggested few directions for improve the system.

The review has written well overall. Please see below comments to address before the next submission.

Major:

1.       Technology: The author should also discuss the various new technology tools available to enhance the learning and bridge the barrier. It may be barrier itself to access the technology.

2.       Often, child may not have been in the center of the decision process and having them in driving seat to set up academic and learning goals, creating environment to fail/growth is paramount to swivel the process in their reach direction. In addition, need for companionship and setting up realistic achievable goals in very important. Please discuss the therapeutic and psychological interventions in this context. If possible, add related literature with success or learning story.

3.       Section 2.3.1 and 2.3.2 are too long and off tracks the reader from the main idea of the review. Please rewrite to include only necessary and relevant information.

4.       The review has some repetitive ideas presented in the different sections. Please proofread to maintain the flow of the content and to be precise to elaborate the point.

Minor:

1.       Type error: line 511, line 547

Author Response

Attached, I provided a response letter that addresses all three reviewers comments. Below, I listed responses to this reviewer's comments. 

Reviewer 1, Comment 1: Technology: The author should also discuss the various new technology tools available to enhance the learning and bridge the barrier. It may be barrier itself to access the technology.

Author Response: We agree that technology can be helpful for enhancing children’s academic functioning but that access to technology presents an additional barrier in many cases. Therefore, we revised the manuscript to include the following section within the current limitations section (section 4.1).

“Prescribed accommodations that depend on families having access to certain technology can contribute to educational barriers. In particular, the education system has increasingly utilized technology to help remediate or compensate for areas of weakness. For example, assistive technology, such as speech-to-text devices and communication devices, has been used to compensate for motor or speech weaknesses. Schools also frequently use computer-based software packages to remediate specific learning difficulties. All of these supports, however, depend on having access to the various technologies. As such, reduced access to such technology can preclude families from receiving the services that they qualify for. This issue has become particularly prevalent in the context of distanced learning due to the COVID-19 pandemic, where many families were expected to provide their own technology to support their children’s schooling. Children in families who experience difficulties accessing technology are, in turn, at risk for being under-served.”   

Reviewer 1, Comment 2: Often, child may not have been in the center of the decision process and having them in driving seat to set up academic and learning goals, creating environment to fail/growth is paramount to swivel the process in their reach direction. In addition, need for companionship and setting up realistic achievable goals in very important. Please discuss the therapeutic and psychological interventions in this context. If possible, add related literature with success or learning story.

Author Response: We thank the reviewer for raising this important topic. We agree that children should be included within the decision-making process and agree with the importance of psychological interventions within this context. To address these points, we have revised the manuscript to include the following paragraph (with relevant literature citations) within the section on current limitations (now section 4.1).

“Evidence suggests that children can effectively achieve self-identified goals (Vroland-Nordstrand et al., 2016) and, thus, it may be appropriate to involve a child with brain injury in the development of their IEP goals. In fact, the self-determination movement has encouraged it (Arndt et al., 2006). However, to date, this movement has not translated from research to the classroom. This problem is attributable to the fact that the IDEA does not require schools to involve students in the development of their IEP; instead, it only requires schools to involve parents to the best of the school's ability. Despite the empirical support and promise for teaching students with disabilities to identify and develop personalized goals, students’ involvement in the IEP process rarely occurs (Arndt et al., 2006; Danneker & Bottge, 2009). It is not until age 16 (age 14 in some states) that IDEA requires an IEP transition plan (often referred to as a student-centered planning meeting) be developed. Even then, though, data from the National Longitudinal Transition Study-2 reported that only 12% of students provided any meaningful input in terms of their interests, needs, and preferences or took a leadership role (Cameto et al., 2004; Shogren & Plotner, 2012).

Reviewer 1, Comment 3: Section 2.3.1 and 2.3.2 are too long and off tracks the reader from the main idea of the review. Please rewrite to include only necessary and relevant information.

Author Response: Per the reviewer’s request, we have significantly reduced the sections on IDEA and Section 504 of 1973 Rehabilitation Act (now labeled sections 3.3.1 and 3.3.2). Specifically, we reduced the word count of those sections by over 50%, from 1,873 to 871.

Reviewer 1, Comment 4: The review has some repetitive ideas presented in the different sections. Please proofread to maintain the flow of the content and to be precise to elaborate the point.

Author Response: We agree with the reviewer that the manuscript would benefit from improving conciseness and cohesiveness. Therefore, we proofread and modified the overall paper to enhance precision, clarity, and continuity of thematic points.

Reviewer 1, Comment 5: Type error: line 511, line 547

Author Response: We thank the reviewer for detecting these errors, which have been corrected within the revised version of the manuscript. 

Reviewer 2 Report

In the manuscript presented by Vanderlind et al., the authors reviews the literature on cognitive, neuropsychological, psychiatric, and academic outcomes of youth with acquired brain injury. They also discussed educational supports, special education law, current limitations within the field, and future directions for solutions. The structure and contents are well organized. The manuscript is very well written and easy to be understood. The conclusion is solid and provides us with clear future directions on solutions for ensuring youth with brain injury be properly identified, kept from being misidentified, and effectively supported in schools. In addition, the language is fluent and precise. However, there are a few issues that need to be addressed before publication.

1.    The authors provide the rates of psychiatric sequelae in children with brain injury, which is good. The prevalence of childhood brain injury and cognitive sequelae after brain injury are also needed.

2.    As introduced in the first part, acquired brain injury in children include TBI and non-traumatic brain injury. The authors discussed the cognitive sequelae, psychiatric sequelae, academic achievement, and need for academic supports in the regard of TBI and non-traumatic brain injury. However, only TBI was discussed in school engagement, moderators of academic outcomes, and current limitations. Non-traumatic brain injury should also be referred in these subsections.  

Author Response

Attached, I provided a letter addressing all three reviewers' comments. Below, I listed responses specific to this reviewer's comments. 

Reviewer 2, Comment 1: The authors provide the rates of psychiatric sequelae in children with brain injury, which is good. The prevalence of childhood brain injury and cognitive sequelae after brain injury are also needed.

Author Response: We appreciate the reviewer’s recommendation to include base rates of the various types of brain injuries and related outcomes. Therefore, we included prevalence rates of pediatric TBI, stroke, meningitis, and near-drowning (one of the primary causes of anoxic brain injuries in youth) within the introduction section. Specifically, we report the following.

“Traumatic brain injury is among the leading cause of morbidity and mortality among youth, with prior work documenting nearly 500,000 annual cases of TBI in the United States among newborns to 14-year-olds (Langlois et al., 2005). Moreover, lifetime estimates of pediatric TBI based on parent report are approximately 2.5%, representing 1.8 million children under the age of 18 years old (Haarbauer-Krupa et al., 2018). Non-traumatic acquired brain injuries are less prevalent than TBI. For example, the yearly incidence of pediatric stroke ranges from 2.5/100,000 (Schoenberg et al., 1978) to 13/100,000 (Lynch et al., 2002), with an average of 6.7/100,000 when combining ischemic and hemorrhagic stroke rates (Deveber 2011). Prevalence rates of bacterial meningitis among febrile infants range from 0.46% to 1.2%, with rates decreasing as age increases (Biondi et al., 2019; Martinez et al., 2015). There is less data on the base rates of pediatric anoxic or hypoxic brain injuries, possibly because of the various causes that result in such injuries. Notably, though, 1 in 301 boys and 1 in 913 girls from birth to 19 years old are hospitalized for near-drowning, a primary cause of these injuries (Wintemute, 1990).”

Additionally, when feasible, we include rates on the provision of educational supports within the academic outcomes section. Specifically, we reference the following findings:

“40% of pediatric stroke patients presented with clinically-significant arithmetic weaknesses.”

“30% of parents of school-aged children with a history of bacterial meningitis report problems with school achievement.”

“A 2008 study reported 25% of youth with TBI having a formal Individualized Education Plan (IEP) or 504 Plan, whereas a 2021 study reported 45% of youth with TBI having an IEP after one year of returning to school.”

“Estimates suggest that nearly 25% of moderate and severe TBI populations are placed in special education classrooms, and nearly 40% repeat a school year or require adaptations.”

“Similar data have been reported in a Swedish population-based study, with 38% of arterial ischemic stroke and 24% of hemorrhagic stroke survivors requiring assistance at school”

“59% of parents of youth with a history of stroke report that their child needs more help than their peers, and in the same study, 19% of pediatric stroke survivors subsequently attended a special school”

Comparatively, there is less data on rates of cognitive difficulties among this patient population. The relatively dearth of prevalence rates, at least in part, stems from the lack of universal definitions of cognitive impairment. Rates of psychiatric sequelae are easier to obtain given their relatively defined diagnostic criteria. Comparatively, most research on neuropsychological outcomes of brain injury identifies areas of difficulty based on a population’s performance relative to normative data or compares different patient populations to identify group differences. For these reasons, we were unable to include base rates within that section.  

Reviewer 2, Comment 2: As introduced in the first part, acquired brain injury in children include TBI and non-traumatic brain injury. The authors discussed the cognitive sequelae, psychiatric sequelae, academic achievement, and need for academic supports in the regard of TBI and non-traumatic brain injury. However, only TBI was discussed in school engagement, moderators of academic outcomes, and current limitations. Non-traumatic brain injury should also be referred in these subsections.

Author Response: We thank the reviewer for raising this important point! As we mention in the manuscript, there is more research on outcomes associated with pediatric TBI than there are as it relates to non-traumatic acquired brain injuries. The discrepancy is, at least in part, due to differences in the base rates of these insults, with pediatric TBI being significantly more common than pediatric stroke, meningitis, and anoxic brain injuries. Thus, the literature reviewed in some sections is more heavily focused on TBI outcomes. In fact, to date, we have been unable to identify any current literature focusing on school engagement (e.g., absenteeism, classroom participation) among samples with histories of non-traumatic acquired brain injuries. We have revised the manuscript to address this point. Specifically, we now say, “the existing literature on this topic has predominantly come from research on TBI; no studies were identified examining school engagement using samples of youth with non-traumatic acquired brain injuries.” Additionally, we have revised the manuscript to increase the focus on non-traumatic acquired brain injury within the moderators of academic outcomes and current limitations sections.

Within the moderators of academic outcomes section, we reference three studies (two on bacterial meningitis and the other on asphyxia) showing that greater severity is associated with worse outcomes. Specifically, we state, “more neurologic complications in the acute stages of bacterial meningitis are linked to poorer academic outcomes, including poorer performance on measures of reading, spelling, and mathematics, across multiple studies. Further, youth with a history of moderate and severe birth asphyxia show poorer reading, spelling, and mathematic abilities than those with mild asphyxia or a healthy comparison group.” We also include a study on pediatric stroke showing, “greater parent-rated executive dysfunction was predictive of poorer performance on spelling and, in particular, math measures.” We further highlight moderators of outcome among pediatric stroke survivors by including two studies focused on age of jury. Specifically, we state, “early age of pediatric stroke has been associated with greater academic challenges. Notably, though, one study showed that children who had a stroke between the ages of one and six years showed relative preservations in subsequent academic functioning (i.e., reading and spelling skills) as compared to those who had a stroke earlier (i.e., younger than one year old) and later (i.e., six to 16 years old) in development.”

Within the current limitations section, we address an issue that is specific to non-traumatic acquired brain injuries – the absence of an appropriate classification category within IDEA. This limitation is a central issue underlying reduced access to educational supports for youth with pediatric stroke, meningitis, and anoxic brain injury histories. We include the following:

“A current limitation of IDEA for children with non-traumatic acquired brain injury is the lack of an appropriate IEP classification category. As mentioned, TBI classification is specific to youth who have an acquired change to their cognitive functioning by means of an external force, which precludes children with histories of stroke, meningitis, anoxic brain injury, etc. As such, children with non-traumatic acquired brain injuries are typically classified under a catchment category, OHI, the same category for that is most commonly used for children with a diagnosis of AD/HD. While classification categories may seem trivial to the unfamiliar reader, they can affect children’s access to certain types of services. For example, in Oregon, youth with TBI have access to a liaison service that helps to bridge the medical and educational settings and support the appropriate sharing of information between sectors. Conversely, those who present with a non-traumatic acquired brain injury are unable to access such services. Thus, inclusion of non-traumatic acquired brain injury within the TBI classification may equalize the playing field for youth with brain injury within the academic setting.”

Additionally, we have reviewed and modified the wording to be more general (referring to brain injury without specification of traumatic vs. non-traumatic brain injury) in sections that are not specific to one type of injury over another.

Reviewer 3 Report

Dear Authors,

I read your work entitled " Back to School: Academic Functioning and Educational Needs 2

Among Youth with Acquired Brain Injury” and here I enclose my recommendations:

1.     This work is full documented, and I congratulate the Authors for the amount of the literature that was included.

2.     The weakness of this work is that we do not see any king of methods on how this literature was selected, what inclusion and exclusion criteria were set.

3.     The Authors do not report what type of review is that. Is it a) Narrative or traditional literature review, b) Critically Appraised Topic (CAT), c) Scoping review, d) Systematic literature review, e) Annotated bibliographies? The authors should clarify this and accordingly to develop this work. If it is none of the type above, then it is considered  like a scientific essay.

Thank you.

Author Response

Attached, I provided a letter addressing all three reviewers' comments. Below, I listed responses specific to this reviewer's comments. 

Reviewer 3, Comment 1: This work is full documented, and I congratulate the Authors for the amount of the literature that was included.

Author Response: We very much appreciate the reviewer’s positive feedback! 

Reviewer 3, Comment 2: The weakness of this work is that we do not see any kind of methods on how this literature was selected, what inclusion and exclusion criteria were set.

Author Response: We agree with the reviewer that the manuscript would benefit from a methods section. Therefore, we have added the following section to the manuscript address this reviewer comment.

“We conducted a narrative or traditional review of neuropsychological, psychiatric, and academic functioning among youth with acquired brain injury. A PubMed and Google Scholar database search was conducted to locate relevant published work. Search terms included pediatric brain injury OR traumatic brain injury OR childhood stroke OR neuro-logical infection OR meningitis OR hypoxic brain injury OR anoxic brain injury AND cognitive OR psychiatric OR academic OR school (re-entry) AND sequelae OR outcomes. Additional literature on special education law was reviewed as well. For select seminal articles, cited referenced and subsequent articles that cited those works were reviewed (i.e., “snowballing”). The authors reviewed eligible articles and excluded articles that were not relevant to the primary topic, case reports, and non-English language studies. A total of 168 articles were included in this review paper. Authors also provided critical and objective analysis on the current state of limitations to educational supports for youth with brain injuries.”

Reviewer 3, Comment 3: The Authors do not report what type of review is that. Is it a) Narrative or traditional literature review, b) Critically Appraised Topic (CAT), c) Scoping review, d) Systematic literature review, e) Annotated bibliographies? The authors should clarify this and accordingly to develop this work. If it is none of the type above, then it is considered like a scientific essay.

Author Response: We thank the reviewer for this suggestion! We have clarified that the review is a narrative or traditional review within the newly-added methods section (see response to Reviewer 3, Comment 2 above).

Round 2

Reviewer 1 Report

Thank you for incorporating the suggested comments. The manuscript is good shape.

Thank you

Reviewer 3 Report

Dear Authors,

I read your work and it is very good that you have addressed all of my suggestions. Please, have a final English editing in the texts added. 

Thank you.